# Formation and Stability of Prebiotically Relevant Vesicular Systems in Terrestrial Geothermal Environments

**DOI:** 10.3390/life7040051

**Published:** 2017-11-30

**Authors:** Manesh Prakash Joshi, Anupam Samanta, Gyana Ranjan Tripathy, Sudha Rajamani

**Affiliations:** 1Department of Biology, Indian Institute of Science Education and Research, Pune 411008, India; manesh.joshi@students.iiserpune.ac.in; 2Department of Earth and Climate Science, Indian Institute of Science Education and Research, Pune 411008, India; anupam.samanta@students.iiserpune.ac.in (A.S.); grtripathy@iiserpune.ac.in (G.R.T.)

**Keywords:** terrestrial geothermal fields, origins of life, protocells, prebiotic compartments, vesicle formation, temperature stability, hot spring, dehydration-rehydration, analogous environments

## Abstract

Terrestrial geothermal fields and oceanic hydrothermal vents are considered as candidate environments for the emergence of life on Earth. Nevertheless, the ionic strength and salinity of oceans present serious limitations for the self-assembly of amphiphiles, a process that is fundamental for the formation of first protocells. Consequently, we systematically characterized the efficiency of amphiphile assembly, and vesicular stability, in terrestrial geothermal environments, both, under simulated laboratory conditions and in hot spring water samples (collected from Ladakh, India, an Astrobiologically relevant site). Combinations of prebiotically pertinent fatty acids and their derivatives were evaluated for the formation of vesicles in aforesaid scenarios. Additionally, the stability of these vesicles was characterized over multiple dehydration-rehydration cycles, at elevated temperatures. Among the combinations that were tested, mixtures of fatty acid and its glycerol derivatives were found to be the most robust, also resulting in vesicles in all of the hot spring waters that were tested. Importantly, these vesicles were stable at high temperatures, and this fatty acid system retained its vesicle forming propensity, even after multiple cycles of dehydration-rehydration. The remaining systems, however, formed vesicles only in bicine buffer. Our results suggest that certain prebiotic compartments would have had a selective advantage in terrestrial geothermal niches. Significantly, our study highlights the importance of validating results that are obtained under ‘buffered’ laboratory conditions, by verifying their plausibility in prebiotically analogous environments.

## 1. Introduction

Potential niches that could have supported the origin of life on the early Earth have long been a topic of study, and that of even intense debate. A given environment needs to fulfill certain criteria in order to be considered as a potential site for the origin of life. To highlight some of the important ones, the environment should have a source of, both, energy and precursor molecules, mechanisms for concentrating these precursors, and the potential for catalyzing energetically uphill polymerization reactions [1]. One such site is marine hydrothermal vents, which led to the popular hypothesis that life on the Earth might have originated and evolved in the oceans [2]. However, it has been argued that the ionic strength and salinity of oceans would have severely limited the facilitation of fundamental processes that are considered to be important for the formation of protocellular entities. These processes mainly include the self-assembly of amphiphiles and the nonenzymatic polymerization of mononucleotides [3]. Additionally, the pH of prebiotic oceans could have been much more acidic than that of the present day oceans because of what has been hypothesized as resulting from elevated levels of CO_2_ in the early Earth atmosphere [4]. This would make it difficult for simple amphiphiles, like fatty acids, to form vesicles as they self-assemble into these higher ordered entities at a pH that is around their pKa (in a range of pH 7 to 9). Furthermore, the problem of dilution of the reactants and products, if left un-sequestered, is an important constraint for this hypothesis [5,6].

An alternative hypothesis, which alleviates the aforementioned issues, is the proposal that life would have originated in terrestrial geothermal fields, such as geysers, hot springs, and tidal pools [7,8,9]. In addition to fulfilling basic criteria for being a plausible hatchery for the beginning of cellular life, their ionic strengths are much lower than that of oceans. This would have made them more amenable for the occurrence and sustenance of prebiotically pertinent processes, including the formation of vesicular membranes by amphiphilic systems. Additionally, the dehydration-rehydration (DH-RH) conditions, which are quite common in these regimes, would have catalysed the nonenzymatic extension of monomers that were capable of encoding information [10,11], and their subsequent encapsulation into membranous compartments to result in the first protocells [12]. Although, it is impossible to know the exact ionic composition of prebiotic oceans and terrestrial geothermal fields, one could take a somewhat top-down approach by systematically studying the aforementioned processes in prebiotically pertinent analogous environments that are found on the extant Earth. This would also provide a more realistic context to discern whether the results that we obtain in the laboratory, under “buffered” conditions, would indeed hold true in realistic natural environments. After all, life started in a natural environment and not in the laboratory.

Towards this extent, an exploration of prebiotically analogous sites was undertaken as a part of an expedition to Ladakh, which is situated in the northern most state of Jammu and Kashmir, in India. This was done under the aegis of the NASA Spaceward Bound program in the August of 2016 [13]. Ladakh is an Astrobiologically relevant site, which is dotted with topological features that are analogous to the landscape of Mars and, hence, it is emulative of early Earth conditions. As part of the geochemistry team, water samples were collected by our group from various sites pertaining to three different hot springs in Ladakh, namely Puga (PU), Chumathang (CH), and Panamic (PA). All of these are alkaline hot springs where the water boils at 85 °C due to the high altitude (3200–4500 m above sea level; Appendix A). It is interesting to note that PU is analogous in nature to the Pilbara hot spring deposits of Western Australia, where the earliest signs of terrestrial life has recently been confirmed [14].

These hot spring water samples provide a prebiotically pertinent backdrop for studying fundamental processes that would have been crucial for the formation of the first protocells. As mentioned earlier, one of these processes would be the self-assembly of prebiotically relevant amphiphiles, such as fatty acids, to form membranous compartments. Fatty acid vesicle formation has been shown to be affected by several parameters, such as temperature, pH, concentration of the fatty acid, ionic strength of the medium, etc. Studies have been performed to understand the effect of single or multiple ions (mostly independently) [3], and that of temperature [15], on the formation and stability of fatty acid vesicles. However, the combined effect of these two crucial factors i.e., ionic composition and temperature, on the aforementioned aspects, is yet to be elucidated. In this study, we set out to address a two-fold question. The first goal was to delineate the effect of multiple ions in varying concentrations, on vesicle formation using different fatty acid systems. We also characterized the stability of these vesicles in niches with varying ionic strengths and at a high temperature. Furthermore, we studied the effect of DH-RH conditions on these systems, as they mimic natural processes (e.g., day-night and wet-dry cycles) that would have played out in prebiotic terrestrial geothermal fields. As a part of the second goal, we undertook studies to delineate the formation and stability of fatty acid vesicles in prebiotic analog conditions, to facilitate comparison between ‘buffered’ versus more ‘realistic’ conditions. We found that the fatty acid and its glycerol derivative was the most robust system among the combinations that were tested. This binary system formed vesicles in all of the hot spring water samples, which had varying ionic strengths. These mixed fatty acid vesicles were also stable at high temperature and retained their propensity to form vesicles under multiple DH-RH cycles. Furthermore, all the combinations of different fatty acids and their derivatives formed vesicles in buffered conditions, but not in the actual hot spring samples, with the exception of the aforementioned binary system. These results clearly indicate that the outcome of experiments that are performed under laboratory conditions are highly context dependent, and, significantly, may not always hold true under natural conditions.

## 2. Materials and Methods

### 2.1. Materials

All fatty acids and their derivatives were purchased from Nu-Chek-Prep (Elysian, MN, USA), except for 1-Decanoyl-rac-glycerol, which was purchased from Sigma-Aldrich (Bengaluru, India). All other reagents and solvents that were used were of the highest commercially available grade.

### 2.2. Methods

#### 2.2.1. Collection of Water Samples from Hot Springs

The water samples were collected from Puga, Chumathang, and Panamic hot springs, from both origin and run-off sites. However, only the samples that were collected from the sites of origin were used for this study. This was because these sites lacked any sort of obvious microbial growth and vegetation, thus reducing concerns of biological contamination and contamination due to human interference. All of the samples were filtered at the site of collection through a 200 nm Whatman syringe filter to avoid microbial contamination, if any, and their temperature and pH were measured. The pH was reanalyzed in the laboratory for all of the water samples that were used in this study (Appendix A).

#### 2.2.2. Formation of Fatty Acid Vesicles

Different concentrations of fatty acids and their derivatives were selected for the experiment (Appendix A), which is similar to what has been previously reported [3,15]. In a typical reaction, the fatty acid alone, or with the pertinent derivative, was first melted by heating above their melting temperature. Appropriate concentrations were then dissolved in chloroform and the chloroform was subsequently evaporated under vacuum to form a fatty acid film. The desired volume of solvent (bicine buffer/hot spring water) was then added to this film to form vesicles. Importantly, the solvent was preheated above the melting temperature of the system. The resultant solution containing vesicles was mixed using a vortex mixer and then visualized under Differential Interference Contrast microscope (AxioImager Z1, Carl Zeiss, Germany), using 40X objective (NA = 0.75).

#### 2.2.3. Dehydration-Rehydration (DH-RH) Experiments

The DH-RH experimental setup was similar to the one that has been used in previous studies [11]. Briefly, a 200 µL solution containing vesicles was taken in a glass vial and was kept on a bench-top heating block, maintained at 75 °C under constant gentle CO_2_ flow. The rehydration was performed using either the same starting solvent (bicine buffer/hot spring water) or with milli-Q water, as indicated appropriately. The duration of each DH-RH cycle was 1 h (Appendix A). Aliquots were taken at intermediate time points and were checked for the presence of vesicles using DIC microscopy. Unless and otherwise mentioned, all of the experiments were performed at least in triplicates.

#### 2.2.4. Geochemical Analysis of Hot Spring Water Samples

The water samples that were collected from the three different hot springs (from their sites of origin) were analysed to discern the major ions that were present in them and to also quantify their concentrations. These measurements were carried out following standard protocols [16]. Briefly, the alkalinity of the samples was measured using an auto-titrator Titrino plus 877 (Metrohm, Switzerland). The concentrations of the major cations (Na^+^, K^+^, Ca^2+^, Mg^2+^ and Li^+^) and anions (Cl^−^, SO_4_^2−^) were measured using ion chromatography instrument Compact IC plus 882 (Metrohm, Switzerland). The accuracy and precision of these analyses were regularly monitored and were found to have average values of ±4%. The net inorganic charge balance (NICB) for these samples were within ±10% (Results Section 3.5), thus ensuring good data quality.

## 3. Results

### 3.1. Formation of Mixed Fatty Acid Vesicles in Hot Spring Water

Analysis of PU, CH, and PA water samples showed that all of them had pH around 8.3 to 8.65 (Appendix A). It is known that certain fatty acids form vesicles in this pH range, and given their importance in the formation of plausible prebiotic compartments, we set out to study whether they could also form vesicles in these hot spring waters. The experiment was performed using different combinations of oleic acid (OA) and its derivatives, oleyl alcohol (OOH), and monoolein (GMO). This mixed fatty acid system was also selected because it has been widely used to study the effect of different parameters on the formation and stability of plausible prebiotic membranes. 

Out of the four combinations of oleic acid and its derivatives, only OA + GMO (6 mM; 2:1 ratio) formed vesicles in the hot spring waters (Figure 1). The number of vesicles that formed qualitatively seemed more in PU followed by CH and PA, possibly owing to the differences in their ionic make up and concentration of the various species therein. Most of the resultant vesicles were large multilamellar or multivesicular vesicles. In the case of PA, a white precipitate was observed on the walls of the reaction container, which dissolved and resulted in vesicles only after prolonged heating. Additionally, in all of the hot spring samples that were analysed, there was always a mixture of vesicles and oil droplets that were observed, with the vesicles staying in the solution phase while the oil droplets adhered to the glass surface of the slide (Please refer to Appendix B for more details about the criteria that was used to distinguish between vesicles and oil droplets). However, in case of 200 mM bicine buffer pH 8.5, which was used as a positive control, vesicles were mainly observed with very few droplets present on some occasions. 

Oleic acid by itself (6 mM), or when present as a mixture of OA + OOH (6 mM; 2:1 ratio), did not form vesicles in the hot spring waters, although they could form vesicles in bicine buffer (Appendix A).

Interestingly, a tertiary system composed of OA + OOH + GMO (6 mM; 4:1:1 ratio) also was unable to form vesicles in the hot spring waters, where only oil droplets were observed (Figure 2 and Appendix A). The addition of alcohol seemed to have a destabilizing effect on vesicles that otherwise could potentially form when only the OA + GMO combination was used. Once again, this effect was observed only in the hot spring waters and not in bicine buffer, wherein the tertiary system readily formed vesicles (Appendix A).

### 3.2. Effect of Chain Length and Saturation on the Vesicle Forming Ability of Fatty Acid and Its Derivatives in the Hot Spring Water Samples

Small chain fatty acids (C8–C12) are thought to be more plausible on the early Earth as they can be synthesized by Fischer-Tropsch-type reaction under hydrothermal conditions [17], and have also shown to be present in meteorites [18,19]. Therefore, the vesicle formation behaviour of a small chain system was also investigated in the hot spring water samples. 10-Undecenoic acid (C11:1) was selected for this experiment to study the effect of reducing the chain length on vesicle formation ability, while keeping the unsaturation in the carbon chain similar to oleic acid and its derivatives. It is important to note that although there are few studies on vesicle formation by 10-Undecenoic acid [20], to our knowledge, this study reports for the first time membrane assembly by this particular fatty acid system in the context of origins of life studies. Results that were obtained were comparable to the oleic acid system, where 10-undecenoic acid (UDA) alone (90 mM), or its combination with undecanoyl alcohol (UDOH) (90 mM; 2:1 ratio), or the tertiary system composed of UDA + UDOH + monoundecanoin (UDG) (90 mM; 4:1:1 ratio), did not form vesicles in the hot spring waters. Only oil droplets were observed in all three of the aforementioned cases in large numbers (Appendix A). The combination of UDA and UDG (90 mM; 2:1 ratio) did form vesicles in PU and CH, as in the oleic acid system, though only oil droplets were initially observed in the PA system. However, prolonged heating of this binary system did induce vesicle formation in PA water (Appendix A). Even in this fatty acid system, all of the four combinations of UDA and its derivatives formed vesicles in 0.2 M bicine buffer pH 8, which served as the positive control (Appendix A). 

Subsequently, the focus was shifted to decanoic acid (DA) and its derivatives. This is one of the most extensively studied small chain amphiphilic system that has long been considered a plausible precursor of prebiotic membranes. Being a saturated fatty acid, this system also allowed for us to examine the effect of saturation on the vesicle formation ability in the hot spring water samples. Once again, as consistent with aforementioned results, DA alone (60 mM), or its combination with decanol (DOH) (60 mM; 2:1 ratio), did not form vesicles in the hot spring waters (Appendix A). However, DA + glycerol derivative of decanoic acid (GMD) (60 mM; 2:1 ratio) readily formed vesicles in all of the hot spring water samples that were tested (Appendix A). Interestingly, the tertiary system comprising of DA + DOH + GMD (60 mM; 4:1:1 ratio) showed peculiar structures in CH and PA, in which the vesicles were typically seen surrounding a central lipid aggregate (indicated by white arrows in Figure 3). This aggregate possibly could be rich in DOH that seems to get excluded from the vesicle forming components of the system, in the form of oil droplet. These structures were not observed in bicine or PU, which showed the presence of regular vesicular structures (top two panels of Figure 3). Table 1 summarizes the vesicle formation results in all the three fatty acid systems that were studied.

### 3.3. Temperature Stability of Vesicles Containing a Mixture of OA and GMO

In all of the vesicle formation experiments detailed above, the solvent (bicine buffer/hot spring water) was preheated to 75 °C before adding to the lipid mixtures. This was to replicate the temperature conditions that were prevalent in the hot spring sample sites, and also to make sure that the temperature of the solvent is well above the melting temperature of the fatty acid system. In order to study the stability of mixed fatty acid vesicles at this high temperature, and in the presence of the multiple ions that are inherent in the samples, the vesicular solutions of OA and GMO (6 mM; 2:1 ratio) were prepared in the respective hot spring waters, and were heated at 75 °C for up to 7 h. The resultant vesicles were reasonably stable at this temperature in, both, the hot spring water samples, as well as in the bicine buffer control (Figure 4 and Appendix A). An interesting result was observed in case of the PA system where prolonged heating led to the dissolution of the starting white precipitate (mentioned earlier), resulting in the formation of large multilamellar vesicles that were observed after 3 h of heating (Appendix A).

### 3.4. Stability of Mixed Fatty Acid Vesicles under Dehydration-Rehydration (DH-RH) Conditions

DH-RH cycles, a prebiotically pertinent and prevalent feature, are quite common in terrestrial geothermal fields and have been shown to facilitate prebiotically important reactions, such as nonenzymatic oligomerization of nucleotides and the encapsulation of polymers into vesicular compartments [10]. Furthermore, to the best of our knowledge, there are no reported studies on the stability of fatty acid vesicles under DH-RH conditions. Therefore, these conditions were simulated in the presence of constant gentle CO_2_ flow and the stability of mixed fatty acid vesicles was checked under these conditions (the early Earth environment has been thought to contain higher amount of CO_2_ [21]).

OA + GMO vesicles (6 mM; 2:1 ratio) were quite stable in 200 mM bicine buffer pH 8.5, even after seven cycles of DH-RH (Figure 5). In PU and CH, the number of vesicles observed decreased with an increasing number of DH-RH cycles, although few vesicles were observed, even after seven DH-RH cycles (Figure 5 and Appendix A). In the case of the PA system, the vesicle solution prepared in PA water was first heated for 3 h to maximize vesicle formation, and subsequently subjected to DH-RH cycles. Vesicles were unstable even after one DH-RH cycle, and small shiny structures were apparent in subsequent cycles (Appendix A). It is pertinent to note that the pH of the bicine buffer control and the hot spring water samples did not change significantly during the DH-RH cycles, excluding the role of pH on the effect that was observed. Furthermore, to check whether the stability of small chain fatty acid vesicles were affected in the hot spring water samples under DH-RH conditions, UDA + UDG vesicles (90 mM; 2:1 ratio) were prepared in PU and were subjected to DH-RH cycles. Vesicles were observed even after seven DH-RH cycles. Additionally, small vesicles tended to fuse together to form larger vesicles as the number of DH-RH cycles increased (Appendix A).

In some of the experiments, rehydration was carried out with 200 mM bicine buffer pH 8.5, having 73.14 mM of [Na^+^], to test the inhibitory effect of high monovalent cation concentrations on vesicle stability. Typically, rehydrating with this bicine buffer would cause a two-fold increase in the amount of Na^+^ ion in the solution, with every cycle of rehydration. Vesicle aggregation was observed after five DH-RH cycles ([Na^+^] = 365.7 mM), (Appendix A). However, no aggregates were observed when rehydration was performed with milli-Q water, in which case the original concentration of Na^+^ remained constant throughout the DH-RH experiment. These results clearly illustrate the inhibitory effect of monovalent cations on vesicle formation when present in high concentrations. 

### 3.5. Geochemical Analysis of the Hot Spring Water Samples

In order to understand the role of ions on the vesicle forming capability of fatty acids, the ionic composition of all the hot spring samples was measured (Table 2). Multiple ions, in varying concentrations, were detected. Overall, the ionic strength of PU was higher than that of CH, followed by PA. The chemistry of the hot spring samples analysed in this study is dominated by monovalent anions (HCO_3_^−^ and Cl^−^) and cations (Na^+^ and K^+^). In these samples, alkalinity accounted for more than 50% of the total anions, whereas Na^+^ and K^+^ concentrations together accounted for more than 90% of the total cations. These chemical patterns are consistent with earlier data reported for hot spring water samples that were collected from these or other near-by locations [22]. Higher concentrations of Na^+^ and K^+^ in these samples confirm the intense weathering of silicate minerals that are present within the basins. Total dissolved solids (TDS) is a measure of chemical weathering intensity in a given basin. It is estimated as the summation of all the major constituents (in mg/L units) available in spring or river water. TDS is generally measured using the chemistry of filtered aliquots of the samples, and, therefore, reflects the total amount of solutes that are present in the hot springs. In the absence of SiO_2_ data, we could not compute the TDS values for the samples investigated herein. However, earlier studies have shown that the average TDS values for thermal springs in these locations vary widely from 550 ± 29 mg/L to 2116 ± 230 mg/L, with the highest concentrations being observed for the samples from Puga [22,23]. This may be because the Puga spring water that is situated close to the Indus Tsangpo Suture Zone has a deeper (5–7 km) source of fluids, which may promote intense chemical weathering [23].

The concentration of major divalent (Mg^2+^ and Ca^2+^) and monovalent (Na^+^ and K^+^) cations, however, were lower (even in PU) than what has been reported to disrupt vesicle formation [3]. In this previous study, these ions had been individually tested for their vesicle disruption ability. However, not much is known about their combined effect on vesicle stability. Therefore, it is possible that these multiple ions (and their varying concentrations) might be causing a combinatorial effect on both vesicle formation and their stability in the hot spring water samples. Interestingly, this effect was seen across all of the fatty acid systems that were tested. Additionally, the possibility of the presence of organic compounds in these hot spring waters cannot be excluded, which could have also affected the vesicle formation process.

## 4. Discussion

Results obtained in this study showed that the mixture of fatty acid and its glycerol derivative was the most resilient system among all of the combinations that were tested. This binary system formed vesicles in all of the hot spring water samples with different ionic strengths. Also, these vesicles were stable at high temperature and under DH-RH conditions. The aforementioned results pertaining to vesicle formation was consistent across all the fatty acid systems that were tested; independent of varying chain length and saturation. The results pertaining to vesicle stability was consistent in both the C18:1 and C11:1 systems, irrespective of the chain length. Furthermore, it is important to note that all of the combinations of different fatty acids and their derivatives formed vesicles in bicine buffer, but only few of them could form vesicles in the actual hot spring water samples.

This study confirms previous observations that the fatty acid and its glycerol derivative seems to be the most robust system among all of the combinations that were tested [3]. Importantly, this binary system also showed a vesicle forming capability under the formidable ‘realistic’ conditions of the hot spring samples that had multiple ions in varying concentrations. Furthermore, they were resilient when subjected to high temperatures and multiple DH-RH cycles. Additionally, these results corroborate the hypothesis that mixed fatty acid vesicles, with a stabilizing counterpart, like glycerol derivative, could have withstood the tough conditions that are encountered in prebiotic environments. The stabilizing effect provided by the glycerol derivative of fatty acid was consistent across all of the fatty acid systems that were tested including the short chained fatty acid, decanoic acid. This fatty acid is long enough to easily form vesicles [24], but also short enough to be found in relatively large abundance under prebiotic conditions, when compared with C18 or C16 systems [18,19]. Therefore, the formation of mixed fatty acid vesicles of DA and its glycerol derivative, under hot spring conditions, indicates that the C10 system could have readily served as membrane components of the first protocells in a terrestrial geothermal field.

Interestingly, when the fatty alcohol counterpart of a particular fatty acid was included in the relevant mixture, there was an inhibitory effect on the vesicle forming capability in certain fatty acid systems. It is important to note that this inhibitory effect from the addition of the alcohol derivative, on the vesicle formation by a tertiary system, was only observed in the hot spring samples (discussed in detail in the Results section). This was not observed in the control reactions where bicine buffer was used to maintain the pH. This result was initially surprising given that previous studies involving the alcohol derivative have demonstrated the formation of stable vesicles with mixed composition [15,25]. Also, vesicle formation by a heterogeneous mixture of fatty acids and their derivatives has been thought to be more prebiotically ‘realistic’ as the soup would have been a composite mixture of diverse types of amphiphiles. However, the significance of our aforementioned result from the fatty alcohol experiments highlights the fact that the stabilizing effect of individual surfactants need not always be additive in all scenarios. The only exception was the DA tertiary system that formed vesicles, even in the hot spring water samples, further strengthening the pertinence of this fatty acid system as plausible prebiotic membranes.

One other significant highlight of this study was to test the stability of mixed fatty acid vesicles at high temperature (prolonged heating) and under DH-RH cycles. It has been reported previously that vesicles made up of fatty acid and their glycerol derivative are thermostable under buffered conditions [15]. In this study, we show that this holds true even under actual hot spring conditions, which further affirms the robustness of this system. Importantly, the stability of amphiphilic systems under DH-RH conditions has not been studied in great detail. There are very few reports on the stability of phospholipid vesicles under DH-RH cycles, in buffered conditions [10,11]. However, given the importance of fatty acids as plausible prebiotic compartments, it is pertinent to study their stability under DH-RH conditions. For the first time, we report that, just like in the case of phospholipid vesicles, the vesicles that are made up of fatty acid and its glycerol derivative continue to retain their vesicle forming capability for up to seven cycles of DH-RH in bicine buffer. However, this ability decreases under hot spring conditions, and also seems to vary across the different hot spring samples.

It is important to understand that the variation in aforementioned stability will be context dependent and will also be affected by the ionic composition (and strength) of the relevant scenario. This is in addition to the temperature of the niche and other crucial parameters that could potentially lead to inhibitory outcomes in certain instances. Towards this, preliminary geochemical analyses of the three hot spring water samples was undertaken to better understand the role of their ionic makeup on, both, the feasibility of vesicle formation and their subsequent stability in these ‘analog’ niches. The chemistry of a hot spring is mainly governed by the relative supply of solutes that result from the weathering of different rocks that are present in the basin. The high Cl^−^ and Na^+^ concentrations in the hot spring water samples that were used in this study, indicate to a supply of these solutes from basinal brines and/or buried evaporite [26]. The remaining Na^+^ in excess of Cl^−^ (i.e., Na* [= (Na^+^) − (Cl^−^)]) in these samples, varies between 6500 µM and 14,310 µM. On an average, the Na* of these samples is about 77 ± 21% of their total Na^+^ concentration. The main source for Na*, K^+^ and Li^+^ in these hot springs is the chemical weathering of silicate minerals. Therefore, a major part of the measured alkalinity must have a silicate origin. Furthermore, the absence of prominent amounts of Mg^2+^ in these samples confirms the insignificant supply of solutes that come from the weathering of dolomites and Mg-rich silicates. The spring water chemistry seems to be, by and large, regulated by weathering of silicate rocks (garnetiferous micaschist, dark phyllites, light quartzite, para gneisses, etc. [22]) that are present in the area. However, it is important to note that the ionic content of a hot spring depends on several factors, like geographical location, type of rock, site of collection, etc. Therefore, the ionic content of hot springs that were analyzed in this study could be different from that of other well studied hot springs, like those in Yellowstone National Park [27].

In conclusion, this work highlights the importance of testing laboratory ‘buffered’ reactions under more ‘realistic’ conditions. Our results clearly indicate how things could dramatically change when one moves from laboratory conditions to more prebiotically pertinent scenarios. All four combinations of fatty acids and their derivatives formed vesicles in bicine buffer controls. However, only the combination of fatty acid and its glycerol derivative could significantly form vesicles in the hot spring water samples. This indicates that terrestrial geothermal fields might have allowed for ‘selection mechanisms’ that would have allowed for the sustenance of certain types of reactions, and mixtures, over others, depending on the physicochemical constraints that are at play in that specific niche. These observations also underline the fact that much work remains to be done to systematically characterize various planetary analogs that might have allowed for (and sustained) chemistries that eventually would have led to the transition from chemistry to biology. Importantly, some of the observations from our stability experiments seem to have implications for discerning what biochemical signatures might survive under a certain set of environmental constraints. This, we believe, should be factored in while evaluating the habitability aspect of any planetary analog. Additionally, a logical extension of this work would be to systematically study the nonenzymatic polymerization of mononucleotides under ‘realistic’ hot spring conditions. Furthermore, it would be interesting and important to characterize how various prebiotically relevant co-solutes, including nucleic acids, amino acids, and other pertinent polymers, might have affected the stability of mixed fatty acid vesicles, under these early Earth and planetary analogous conditions.

## Figures and Tables

**Figure 1 life-07-00051-f001:**
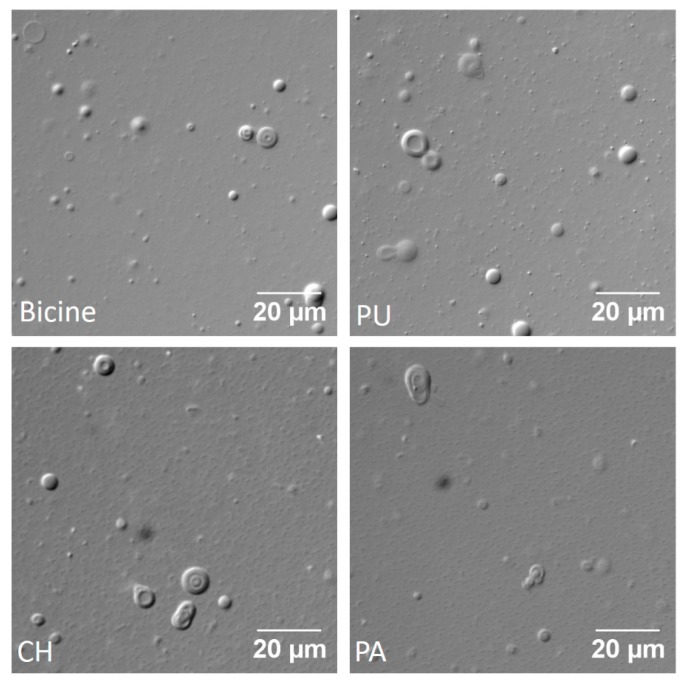
Oleic acid (OA) + monoolein (GMO) system (6 mM; 2:1 ratio): Vesicles formed in 200 mM bicine buffer as well as in all the hot spring water samples i.e., in Puga (PU, top right), Chumathang (CH, bottom left) and Panamic (PA, bottom right), respectively, as shown in the above figure.

**Figure 2 life-07-00051-f002:**
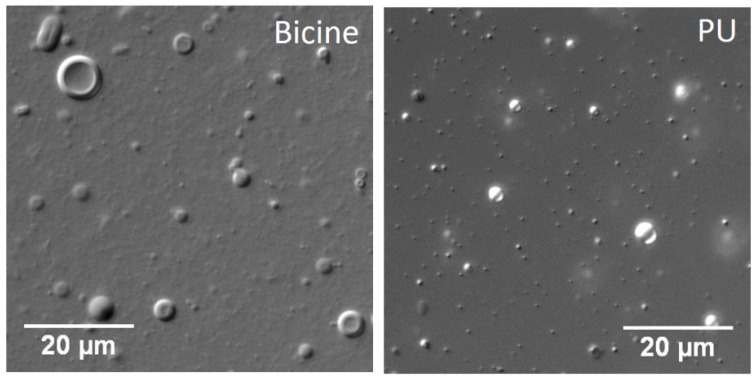
Combination of OA, oleyl alcohol (OOH) and GMO (6 mM; 4:1:1 ratio) formed vesicles in 200 mM bicine buffer pH 8.5 but not in Puga (PU).

**Figure 3 life-07-00051-f003:**
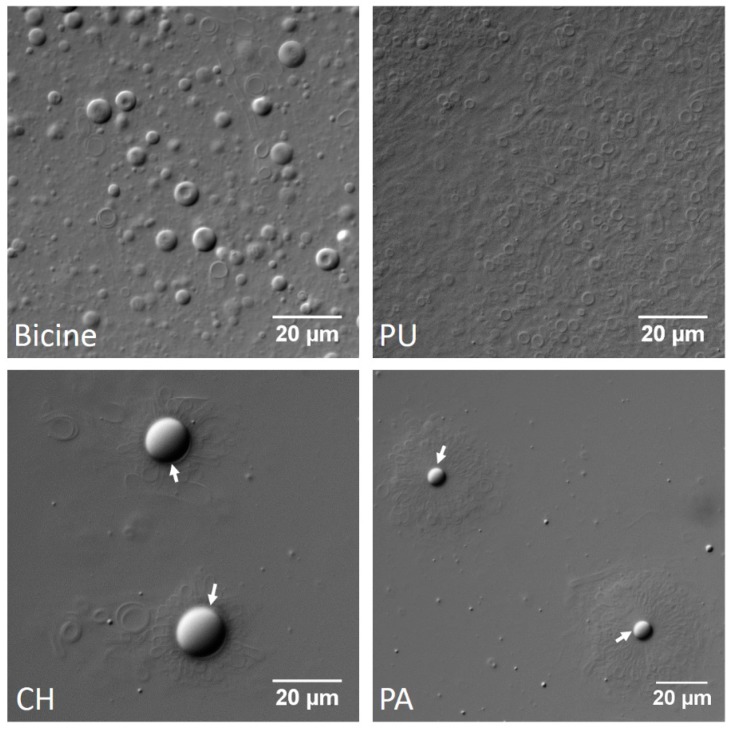
Combination of decanoic acid (DA), decanol (DOH) and glycerol derivative of decanoic acid (GMD) (60 mM; 4:1:1 ratio) formed vesicles in 200 mM bicine buffer pH 8 and also in Puga (PU), as indicated in the top two panels. Interesting structures were observed in the Chumathang (CH) and Panamic (PA) systems (bottom two panels), in which vesicles were present surrounding a central lipid aggregate (indicated by arrows in the respective panels).

**Figure 4 life-07-00051-f004:**
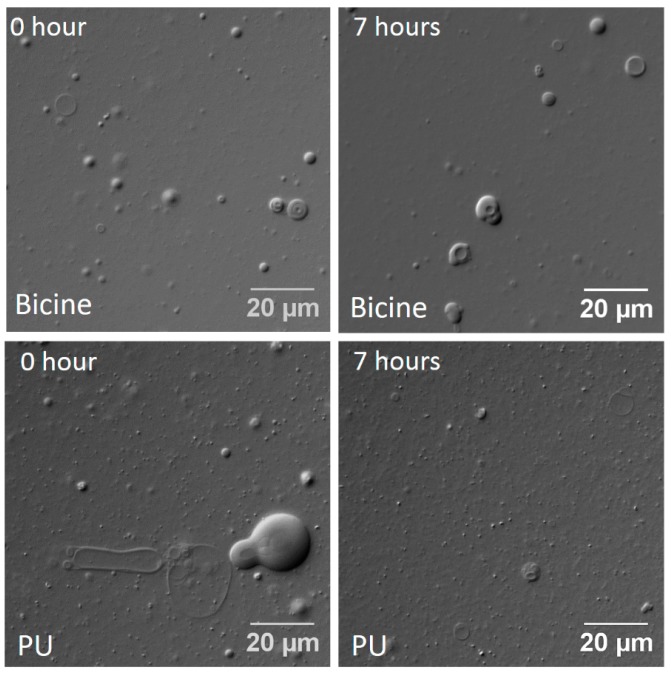
Temperature stability of OA and GMO (6 mM; 2:1 ratio) mixed vesicles in 200 mM bicine buffer pH 8.5 (top two panels) and in Puga (PU, bottom two panels). Vesicles were observed in both the systems, albeit to varying extent, even after heating the reaction samples at 75 °C for 7 h.

**Figure 5 life-07-00051-f005:**
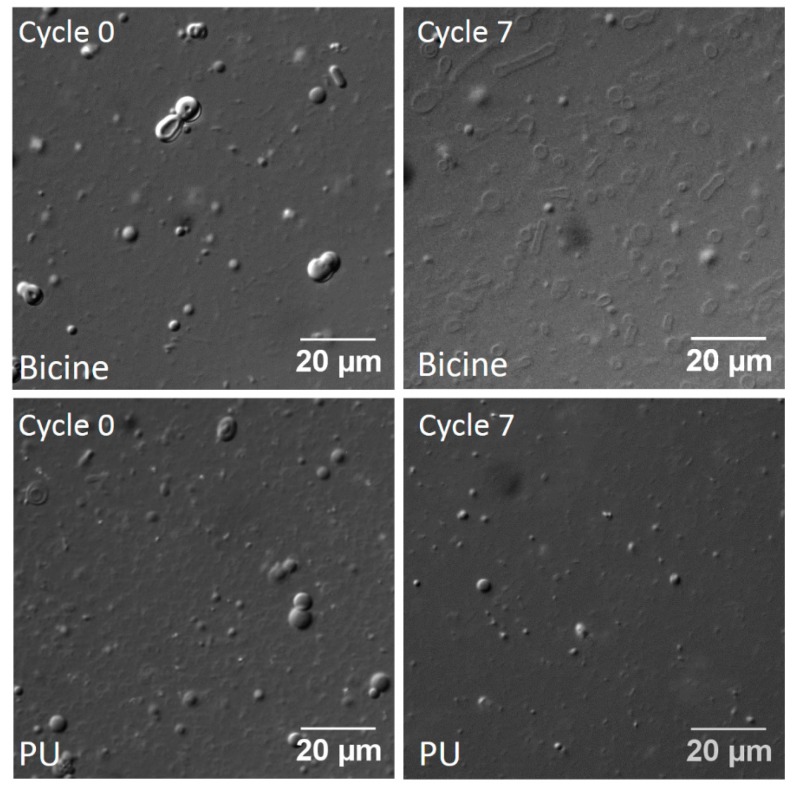
Stability of OA and GMO mixed vesicles (6 mM; 2:1 ratio) under DH-RH conditions in 200 mM bicine buffer pH 8.5 (top two panels) and in Puga (PU, bottom two panels).

**Table 1 life-07-00051-t001:** Vesicle formation experiment results in 0.2 M bicine buffer and Puga (**A**), Chumathang and Panamic (**B**) water samples that were tested using three different fatty acid systems, as indicated. Presence of vesicles is denoted by (✓) while (✗) denote their absence. The pH of bicine buffer control was 8.5 for oleic acid (OA) system and 8 for undecenoic acid (UDA) and decanoic acid (DA) systems.

**A**						
**Combination of Fatty Acid and Its Derivatives**	**Presence or Absence of Vesicles in the Various Fatty Acid Systems Studied**					
	**0.2 M Bicine Buffer**			**Puga**		
	**DA (C10:0) System**	**UDA (C11:1) System**	**OA (C18:1) System**	**DA (C10:0) System**	**UDA (C11:1) System**	**OA (C18:1) System**
only acid	✓	✓	✓	✗	✗	✗
acid + alcohol (2:1 ratio)	✓	✓	✓	✗	✗	✗
acid + glyceride (2:1 ratio)	✓	✓	✓	✓	✓	✓
acid + alcohol + glyceride (4:1:1 ratio)	✓	✓	✓	✓	✗	✗
**B**						
**Combination of Fatty Acid and Its Derivatives**	**Presence or Absence of Vesicles in the Various Fatty Acid Systems Studied**					
	**Chumathang**			**Panamic**		
	**DA (C10:0) System**	**UDA (C11:1) System**	**OA (C18:1) System**	**DA (C10:0) System**	**UDA (C11:1) System**	**OA (C18:1) System**
only acid	✗	✗	✗	✗	✗	✗
acid + alcohol (2:1 ratio)	✗	✗	✗	✗	✗	✗
acid + glyceride (2:1 ratio)	✓	✓	✓	✓	✓ ^#^	✓ ^#^
acid + alcohol + glyceride (4:1:1 ratio)	✓	✗	✗	✓	✗	✗

^#^ Vesicles formed after prolonged heating.

**Table 2 life-07-00051-t002:** Geochemical analysis of hot spring water samples under study. The overall ionic strength decreases from Puga to Panamic. The net inorganic charge balance (TZ^+^/TZ^−^) around 1 indicates the high quality of analysis.

Hot Spring Water	pH	Major Cations					Major Anions			TZ^+^	TZ^−^	TZ^+^/TZ^−^
		Na^+^	K^+^	Ca^2+^	Mg^2+^	Li^+^	HCO_3_^−^	Cl^−^	SO_4_^2−^			
		All Values in mM								µE		
Puga	8.48	26.14	2.08	0.89	0	1.26	14.57	11.83	1.27	31258	28936	1.08
Chumathang	8.64	15.54	0.55	0.22	0.005	0.44	10.2	2.99	2.53	16980	18248	0.93
Panamic	8.37	6.76	0.14	0.36	0.002	0.05	5.68	0.26	0.99	7665	7918	0.97

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
