# Peer review of "Formation and Stability of Prebiotically Relevant Vesicular Systems in Terrestrial Geothermal Environments"

_life, 2017, doi:10.3390/life7040051_

Round 1

Reviewer 1 Report

The manuscript presented by Dr. Rajamani and collaborators focuses on the formation and the stability of allegedly prebiotic vesicles under simulated hydro-geo-chemical conditions (based on real water samples from a geothermal Indian site) and in laboratory-buffered conditions.

Firstly it should be mentioned the importance of verifying the range of existence of vesicles/protocells in hydro-geo-chemical conditions that simulate the primitive Earth in several environments (marine, fresh water, lagoon, hydrothermal vents, ...). In this context, the study of Rajamani et al. contributes significantly to the ongoing research on protocellular structures.

This study nicely demonstrate that not only the membrane components (fatty acid, fatty alcohol, acylated glycerols) determine the formation of lamellar phases, their stability, etc., but also the environment. Counter ions like Na, K, Mg, Ca, and their complementary anions carbonate, chloride, sulfates tune, in complex way, the self-assembly of amphiphilic compounds.

In a way, this study makes a step further with respect to previous knowledge where only lipid diversity was tested. Here also environmental diversity is tested. A very intriguing scenario emerges.

For these reasons, I certainly support the publication of this work in Life.

However, there are also some critical aspects that need to be considered before fully accept the manuscript.

The presentation style can be improved. Several times, while reading, I feel that I missed some information (pH, concentration, ionic strength, composition of the membrane), that I found later in the text, but the first impression - as reader - is that I missed something. This is a feeling that I had in several parts of the manuscript. Just to give an example, in line 161-162 (page 4), the Authors write "possibly owing to the differences in their ionic make up and concentration of the various species therein". Of course there will be differences, but it is not written what is the difference. The reader has to go to the table (page 11), but pH is not in the Table. This makes the text not so "fluent" to be read. I suggest to mention soon the composition/pH of the different types of hot-spring-water (HSW), the composition of the membranes and any other info that help the readers to see exactly what is going on while reading the results.
Another example, when speaking of DH-RH cycles, one discovers that experiments were done by re-hydrating with buffer or with water. But it is not clear why samples were re-hydrated with buffer (scope of the experiment).

Major Revisions
-----------------------
1. When discussing the ocean features, emphasize the ionic strength and the salinity but does not comment on the pH of ocean. Some authors have analyzed the problem (J. Kua and J. L. Bada, Origins Life Evol. Biosphere, 2011, 41, 553-558), but the authors did not mention this important parameter.

2. On line 132 it is described a constant CO2 flow. Did this change the pH of the aqueous phase?

3. Line 221 and line 284 (Table 1 footnote), and elsewhere. In many cases, vesicles are formed upon heating. Are they stable after cooling? For how long?

4. I am not sure to understand what the Authors mean by "stability" under DH-RH. Probably the vesicles break under DH, if all water evaporate. Under RH conditions, vesicles re-form from bilayer fragments. Please specify better what is meant by "stability" in DH-RH conditions.

5. The meaning of total dissolved solid should be explained, as the manuscript is probably of interest of researchers not fully familiar with terminology of water analysis. Moreover, as the samples were filtered before use, please specify whether TDS is a measure linked to the soluble part (filtrate after dehydration) or not.

6. line 425: "stabilizing effect of glycerol.." maybe better as "stabilizing effect of linking a glycerol to fatty acid ..." or "liked glycerol.."

7. An important technical point in evaluating what kind of compartment is formed when lipids are dispersed in aqueous phase is the capability, by the authors, of distinguish between vesicles (lamellar phase) and lipid droplets. The authors did not comment in details what kind of objective criteria followed for doing this, and how interested readers can learn to do the same. Generally, vesicles are easily spotted when fluorescent compounds are entrapped in their lumen. But in this study transmitted light microscopy was applied, with a DIC filter. Whereas in some case the differences are very evident (see for example Fig. S15), in other cases the images are not very easy to decipher. Please explain and convince the readers that conclusions dranw in this paper are robust with respect to identification of vesicles/droplets by DIC microscopy. A preliminary analysis of systems that certainly form vesicles or droplets might be useful for example.

Minor Revisions

--------------------------
1. On line 69, please give an idea of the pH (also line 114). Similarly, an idea about fatty acids and derivatives concentration would be useful for the readers (e.g. line 117)

2. To be consistent, if oleyl alcohol is OOH, undecenoyl alcohol should be UDOH not UDOL. (similarly DOL --> DOH) But the authors are free of course to use the abbrevations that they like more, as they are non-standard abbrevations.

3. line 236: how the authors conclude that the droplets are DOL-rich?

4. Chapter 5. Very probably, for a better reading, such section would be best placed at the beginning of the manuscript - but this is just a suggestion

5. line 430, maybe add at the end of the sentence "... of the first protocells in a terrestrial geothermal field."

6. On lines 334-335, authors state that no study has been reported on stability of fatty acid systems under DH-RH conditions. I am not fully sure, but probably Deamer/Monnard and colleagues have investigated several systems. Please thoroughly check in their publications.

Author Response

We thank Reviewer 2 for their detailed and critical analysis of the manuscript. The recommendations, questions and suggestions have surely helped us improve the readability and comprehensiveness of the article.

We have systematically addressed all of the comments raised by Reviewer 2 and have detailed it point wise below.

I. The presentation style can be improved. Several times, while reading, I feel that I missed some information (pH, concentration, ionic strength, composition of the membrane), that I found later in the text, but the first impression - as reader - is that I missed something. This is a feeling that I had in several parts of the manuscript. Just to give an example, in line 161-162 (page 4), the Authors write "possibly owing to the differences in their ionic make up and concentration of the various species therein".
Another example, when speaking of DH-RH cycles, one discovers that experiments were done by re-hydrating with buffer or with water. But it is not clear why samples were re-hydrated with buffer (scope of the experiment).

The presentation style has been improved by including appropriate information in requisite places so as to make sure that the reader has all the necessary information at hand while reading. Furthermore, the reason for using buffer as the rehydrating agent in some of the DH-RH experiments has been mentioned in the revised manuscript. 

II. Major Revisions
-----------------------
1. When discussing the ocean features, emphasize the ionic strength and the salinity but does not comment on the pH of ocean. Some authors have analyzed the problem (J. Kua and J. L. Bada, Origins Life Evol. Biosphere, 2011, 41, 553-558), but the authors did not mention this important parameter.

The details pertaining to the pH of prebiotic oceans has been added in the introduction section (lines 45-49) along with the inclusion of the aforementioned reference.

2. On line 132 it is described a constant CO2 flow. Did this change the pH of the aqueous phase?

As mentioned in line 138 of the revised manuscript, the CO2 flow was gentle and did not significantly alter the pH of reaction mixtures during the course of the experiment. We observed no change in the pH of 200 mM bicine buffer, which remained 8.5 throughout the DH-RH experiment. Also, the pH of the hot spring water samples decreased by 0.5-1.0 pH unit after addition of fatty acid(s) at the beginning of the experiment. However, this pH remained constant during the rest of the DH-RH cycles. It is possible that the bicarbonate and silicate (not measured in our analysis) that might be present in these hot spring samples are providing a buffering effect.

3. Line 221 and line 284 (Table 1 footnote), and elsewhere. In many cases, vesicles are formed upon heating. Are they stable after cooling? For how long?

We observed a white precipitate (possibly of vesicles) on the wall of the reaction tube only in the case of Panamic water sample (PA). This precipitate dissolved after 3 hours of heating at 75°C, resulting in large sized vesicles.Additionally, after dissolution, these vesicles were stable at lower temperature as we performed all the microscopic imaging at 18°C and continued to observe the vesicles.

Furthermore, during some of the vesicle formation experiments, the vesicular solution of PA samples was stored at 4°C for 48 hours. On microscopic analysis, vesicles were observed even after 48 hours. This solution was gently heated before microscopy in order to raise its temperature above the phase transition temperature of the respective fatty acid system.

4. I am not sure to understand what the Authors mean by "stability" under DH-RH. Probably the vesicles break under DH, if all water evaporate. Under RH conditions, vesicles re-form from bilayer fragments. Please specify better what is meant by "stability" in DH-RH conditions.

Dehydration causes all water to evaporate, leading the vesicles to fuse and flatten out to form multilamellar matrices, and on subsequent rehydration, vesicles spontaneously bud out from these matrices (reference 8 in the revised manuscript). What we mean by “stability” in DH-RH conditions is the ability of the amphiphilic systems to keep reforming vesicles even after multiple rounds of DH-RH cycling. This characteristic is especially important for the encapsulation of plausible prebiotic genetic polymers that could get synthesized during these DH-RH cycles (reference 10 in the revised manuscript). We have added a phrase in the abstracts section and made appropriate changes in other sections of the manuscript to reflect the aforementioned clarification. 

5. The meaning of total dissolved solid should be explained, as the manuscript is probably of interest of researchers not fully familiar with terminology of water analysis. Moreover, as the samples were filtered before use, please specify whether TDS is a measure linked to the soluble part (filtrate after dehydration) or not.

Details pertaining to TDS have been included in the revised manuscript (lines 393-402).

6. line 425: "stabilizing effect of glycerol.." maybe better as "stabilizing effect of linking a glycerol to fatty acid ..." or "liked glycerol.."

Necessary changes have been made to reflect the above in the revised manuscript (lines 432-433).

7. An important technical point in evaluating what kind of compartment is formed when lipids are dispersed in aqueous phase is the capability, by the authors, of distinguish between vesicles (lamellar phase) and lipid droplets. The authors did not comment in details what kind of objective criteria followed for doing this, and how interested readers can learn to do the same. Generally, vesicles are easily spotted when fluorescent compounds are entrapped in their lumen. But in this study transmitted light microscopy was applied, with a DIC filter. Whereas in some case the differences are very evident (see for example Fig. S15), in other cases the images are not very easy to decipher. Please explain and convince the readers that conclusions dranw in this paper are robust with respect to identification of vesicles/droplets by DIC microscopy. A preliminary analysis of systems that certainly form vesicles or droplets might be useful for example.

A separate appendix has been included in the revised manuscript (lines 529-551), which discusses in detail the criteria that we used to differentiate between vesicles and oil droplets.

III. Minor Revisions

--------------------------
1. On line 69, please give an idea of the pH (also line 114). Similarly, an idea about fatty acids and derivatives concentration would be useful for the readers (e.g. line 117)

We had collected hot spring water samples from various sites pertaining to three different hot spring systems in Ladakh, namely Puga (PU), Chumathang (CH) and Panamic (PA). Though these systems are alkaline in nature, the pH of the water tends to vary depending on the site of collection and also the season during which the sample is collected. Since the introduction section involved an overview of the various hot spring water samples that were collected, we chose to specify in, both, the methods and results sections the pH of those hot spring water samples that we finally worked with. However, we do know from previous reports that these hot springs are generally alkaline in nature (reference 22 and 23 in the revised manuscript).

Furthermore, the details of the pH of the hot spring water samples and the concentrations of fatty acids and their derivatives were included as supplementary tables for the sake of brevity because there was a lot of relevant information that needed to be included for the readers.

2. To be consistent, if oleyl alcohol is OOH, undecenoyl alcohol should be UDOH not UDOL. (similarly DOL --> DOH) But the authors are free of course to use the abbrevations that they like more, as they are non-standard abbrevations. 

The abbreviations have been changed to reflect the reviewer’s comment on keeping things consistent.

3. line 236: how the authors conclude that the droplets are DOL-rich? 

In reference number 3, the authors mention “Oil droplets of DOH were detected even at low salt concentrations owing to the desorption of DOH out of the mixed composition membranes.” We suspect that, in our tertiary system experiment (DA + DOL + GMD) something similar is happening, as the authors of reference 3 had also used decanoic acid system for their experiments. However, we are not sure as to why these structures do not appear in oleic acid or 10-undecanoic acid tertiary systems.

4. Chapter 5. Very probably, for a better reading, such section would be best placed at the beginning of the manuscript - but this is just a suggestion

Thank you for this suggestion. We did consider this but chose to include it where we eventually did for two reasons. Firstly, we did not want the readers to prematurely assume that this paper was geochemistry heavy, which it is not. Secondly, the flow of our manuscript reflects the chronology of experiments that were undertaken. We do believe that this flow is more appropriate for this article.

5. line 430, maybe add at the end of the sentence "... of the first protocells in a terrestrial geothermal field."

Necessary changes have been made in the revised manuscript.

6. On lines 334-335, authors state that no study has been reported on stability of fatty acid systems under DH-RH conditions. I am not fully sure, but probably Deamer/Monnard and colleagues have investigated several systems. Please thoroughly check in their publications.

As mentioned in the discussion section (lines 461-464), David Deamer and colleagues, and also in previous work from our lab, the stability of phospholipid vesicles under DH-RH conditions were studied (reference number 10 and 11 in the revised manuscript). However, to the best of our knowledge, there are no reported studies on the more prebiotically relevant stability of fatty acid vesicles under DH-RH conditions.

Reviewer 2 Report

I have to admit that when I read the abstract, I had low hopes hopes about this paper, because I assumed that experiments done with samples from hot springs was going to be heavy on the gimmick side of science. However, I was pleasantly surprised to see that instead, a nice, useful study was performed and presented. The comparisons with laboratory experiments provide good controls. I would have expected that the fatty alcohols would have helped more in spring water. The data make the convincing case for the importance of the glycerol monoesters in stabilizing vesicles, even the short 10 carbon systems. The science is sound and the presentation clear.

For me, the article can be published as is. I do wonder how the measured salt concentrations compare to other hot spring systems that have been studied before.

Author Response

We thank Reviewer 1 for their encouraging and supportive remarks pertaining to our work. The reviewer wondered about the following “I do wonder how the measured salt concentrations compare to other hot spring systems that have been studied before.” 

The salt concentrations of various hot springs is contingent on a whole host of factors including the nature of rock in the environment, clay etc. To reflect this, we have included a reference from US Geological Survey (Ref no. 27) that involved a detailed preliminary report wherein geochemical analyses of various Yellow Stone hot springs was carried out between1980 and1993 (lines 487-490). 

Reviewer 3 Report

Title: Formation and stability of prebiotic vesicular systems in terrestrial geothermal environments
Authors: Manesh Prakash Joshi, Anupam Samanta, Gyana Ranjan Tripathy, Sudha Rajamani *

To give adequate information to the authors to ameliorate the quality of this paper is very hard. The english written is ok, but the background and the contest of the reserach carried out by Rajamani and co-workers seem repeat the work done by others (i.e. Ref 13). The experimental part is not exhaustive and the results seems to be incomplete from different points of view.

A general remarque is that the article lacks of a certain "energy", and impact in the field of prebiotic chemistry of amphiphiles. The author's best result is to use water sample from hot springs and to hydrate mixtures of lipids, fatty acids and other lipids precursors, thus, to observe the stability of these membranes in DH-RH cycles at elevated temperature.

This part of the work is not a novelty. 

The paper lacks of a figure in witch the structure of the different amphiphiles are elucidated and progressive numbers were attribute to each structure and used in the text. The use of acronyms like OA, OOH should make mistrunderstanding in each type of reader from chemist to bioligist and geologist as well.

The authors do not specify if the analysis of the anions and cations of the water from hydrotermal springs was carried out also before the starting of their experiments and that their results are a novelty or only a second analysis of water already done by others.

Odd numbers of prebiotic plausibly alcohols or related compounds such as uncedyl-phosphates and undecyl phosphatidyl ethanol amines were recently synthesized under prebiotic plausibly conditions. The hydration of those compounds, alone or in mixtures in buffers that reply the water of hydrotermal vents was already proven and compared to those of "buffered" solution of simply milliQ water. Hovewer, C11:1 acid represent a novelty in the field and is one major evidence of a small novelty in the field but not enought, and its plausible presence in a prebiotic scenario is questionable.

Chemical analysis of the vesciles after the hydratation is not reported and is one major drawback.

In general, the figure legend is poor and is preferable a more detaliled one, including pH values, buffer type and so on...

Furthermore, this referee has a question. Do the used water was sterilized? Doit was treated to avoid the presence of any organic compounds? The fact that the authors used a "natural buffer" does not means that this solution is itself free of "plausible" amphiphiles that should be present in a natural enviroment like a lake, sea water and so on.

Again, the authors after hotspring water analysis do not tried to "reproduce" this water in a laboratory starting from milliQ water and dissolving appropriate concentration of salts.

Conclusion: I do not reccomend the paper for publication in MDPI Life journal.   

Author Response

We wish to mention that we humbly choose to not agree with a lot of aspects that Reviewer 3 has mentioned in their review. We have systematically addressed this in the below points.

1. The english written is ok, but the background and the contest of the reserach carried out by Rajamani and co-workers seem repeat the work done by others (i.e. Ref 13). The experimental part is not exhaustive and the results seems to be incomplete from different points of view.

Reference 13 (which is reference 14 in the revised manuscript), is a geology heavy paper, where authors look for microbial biosignatures in the hot spring deposits from Pilbara, Australia. However, and importantly, our study involves a biochemical approach wherein we systematically analysed the formation and stability of plausible prebiotic vesicular systems under natural, prebiotically analogous conditions. We absolutely do not see any similarity between our work and that of authors from reference 13 (which is reference 14 in the revised manuscript).

2. A general remarque is that the article lacks of a certain "energy", and impact in the field of prebiotic chemistry of amphiphiles. The author's best result is to use water sample from hot springs and to hydrate mixtures of lipids, fatty acids and other lipids precursors, thus, to observe the stability of these membranes in DH-RH cycles at elevated temperature.This part of the work is not a novelty. 

We once again choose to humbly disagree with Reviewer 3’s comment on the “lack of a certain energy” in this manuscript. If anything, we, along with the Spaceward Bound expedition team, would like to assure the Managing Editor that this whole process, right from the collection of hot spring water samples, to experimentally characterizing the stability of various fatty acid systems that eventually culminated in this manuscript, has been a journey driven by a lot of positive energy, enthusiasm and dedication.

Importantly, this type of work has not been reported previously to the best of our knowledge. Furthermore, the fact that Life chose to go with a special issue on "Hydrothermal Vents or Hydrothermal Fields: Challenging Paradigms", is indicative of the importance and relevance of our kind of work.

3. The paper lacks of a figure in witch the structure of the different amphiphiles are elucidated and progressive numbers were attribute to each structure and used in the text.

Structures of different fatty acids and their derivatives, and their abbreviations, have been included in the supplementary material of the revised manuscript.

4. The use of acronyms like OA, OOH should make mistrunderstanding in each type of reader from chemist to bioligist and geologist as well.

These acronyms are very commonly used in the field (please refer to reference numbers 3, 15, 25 in the revised manuscript). The intent of using these acronyms for fatty acids and their derivatives was to maintain this nomenclature and, also, to avoid the repetition of the same word multiple times in the manuscript.

5. The authors do not specify if the analysis of the anions and cations of the water from hydrotermal springs was carried out also before the starting of their experiments and that their results are a novelty or only a second analysis of water already done by others.

The order of the various sub-headings in the Results section of the manuscript is reflective of the chronological order in which experiments were conceived and performed. The results from our experiments that looked at vesicle formation and stability of various fatty acid systems indicated to us that the ionic content of the hot spring samples might be potentially playing a role in the results that we obtained. We, therefore, decided to carry out the geochemical analysis of the hot spring water samples that we used in our experiments in collaboration with an established geochemist, Dr. Tripathy (who is a co-author on this manuscript). The geochemical analysis data in Table 2 of the manuscript is for the water samples that we specifically used in our experiments. In the past, studies have been performed to characterize the geochemical nature of hot spring samples from similar locations. However, the concentration of anions and cations in these niches depends on several factors including the exact location from where the sample was sourced, to the time of year it was collected etc. We, therefore, wish to highlight that our analysis is not just a secondary type of analysis that has been previously reported. Importantly, our analysis is directly pertinent to our specific observations.

6. Odd numbers of prebiotic plausibly alcohols or related compounds such as uncedyl-phosphates and undecyl phosphatidyl ethanol amines were recently synthesized under prebiotic plausibly conditions. The hydration of those compounds, alone or in mixtures in buffers that reply the water of hydrotermal vents was already proven and compared to those of "buffered" solution of simply milliQ water. Hovewer, C11:1 acid represent a novelty in the field and is one major evidence of a small novelty in the field but not enought, and its plausible presence in a prebiotic scenario is questionable.

To the best of our knowledge, we have not come across this reference and would like to know more about this study.

7. Chemical analysis of the vesciles after the hydratation is not reported and is one major drawback.

Previous studies have shown that fatty acids are stable over a range of pH and also at high temperatures (reference no 3 and 15 in the revised manuscript). To check if this was the case in our studies, we did carry out preliminary TLC analysis of the fatty acid systems in question. For most part, the system’s components survived multiple DH-RH cycles in buffered conditions. This was somewhat different in the case of the hot spring water based samples. In the hot spring waters, the propensity to form large number of vesicles seemed to reduce, which was clearly reflected in our systematic microscopic analysis (Fig 5, Fig S13 and Fig S14 in the revised manuscript).

8. In general, the figure legend is poor and is preferable a more detaliled one, including pH values, buffer type and so on...

We would like to confirm that we have mentioned the type of buffer, its concentration and pH, wherever appropriate, both, in the figure legend and also in the main text. Furthermore, we preferred to not mention the pH of hot spring water samples multiple times to avoid repetition. 

9. Furthermore, this referee has a question. Do the used water was sterilized? Doit was treated to avoid the presence of any organic compounds? The fact that the authors used a "natural buffer" does not means that this solution is itself free of "plausible" amphiphiles that should be present in a natural enviroment like a lake, sea water and so on.

The hot spring water samples were collected from sites which seemed to be subjected to, predominantly, geochemical processes (as evaluated by the geochemistry experts in our scientific team who were part of the SBI Ladakh expedition). Importantly, these sites were devoid of visible microbial contamination as has been discussed in the Methods section. However, to be extra-cautious, the spring waters collected were subjected to filtration using a 0.2 uM filter right at the point of collection. Additionally, our TLC analysis of the water samples did not indicate the presence of other amphiphiles in the system.

10. Again, the authors after hotspring water analysis do not tried to "reproduce" this water in a laboratory starting from milliQ water and dissolving appropriate concentration of salts.

We did try mimicking the ionic content of the hot spring waters in the lab, starting with milli-Q water as the solvent. However, it was very difficult to maintain a constant pH over the course of the experiment as there is no buffering agent in the milli-Q water. To give an example, we started with only milli-Q water with pH adjusted to 8.4 using NaOH. After addition of the fatty acid, the pH reduces to 6.5 and this pH does not support the formation of vesicles from pure oleic acid system (even at 6mM concentration that is way above the CVC of this system). We, however, did see some vesicles when we used mixed fatty acid system of oleic acid and its glycerol derivative (6 mM; 2:1 ratio). This, for us, strengthens the fact that glycerol derivative provides a stabilizing effect to the fatty acid system and this has been discussed in detail in our manuscript. Furthermore, the “instable pH problem” becomes compounded when salts of different acids and bases are added to the milli-Q water, as each salt has its own effect on the pH. Additionally, there could be other ions in the hot spring samples that are present in really trace concentrations, which might have gone undetected in our analysis. The combination of all the major and minor ions present could potentially have a combined effect on the formation and stability of fatty acid vesicles in the hot spring samples. For these reasons, it is non-trivial to test the individual effect of all the different major and minor ions that are present in these hot spring water samples by mimicking their concentrations in milli-Q water.

Round 2

Reviewer 1 Report

The Authors replied to all my observations and changed the manuscript correspondingly. My suggestion is to accept the paper in the present form

Reviewer 3 Report

I do appreciate the efforts done by the authors for the preparation of this novel version of the manuscript, that results better respect to the first (and non acceptable) submission.